# Effect of Intermittent and Mild Cold Stimulation on the Immune Function of Bursa in Broilers

**DOI:** 10.3390/ani10081275

**Published:** 2020-07-26

**Authors:** Yanhong Liu, Ge Xue, Shuang Li, Yajie Fu, Jingwen Yin, Runxiang Zhang, Jianhong Li

**Affiliations:** 1College of Life Science, Northeast Agricultural University, Harbin 150030, China; lyh_tre@126.com (Y.L.); xuege19961226@163.com (G.X.); 15804639630@163.com (S.L.); mengtian1127@163.com (Y.F.); YinJingwen27@163.com (J.Y.); 2Key Laboratory of Chicken Genetics and Breeding, Ministry of Agriculture and Rural, College of Animal Science and Technology, Northeast Agricultural University, Harbin 150030, China

**Keywords:** broilers, cold stimulation, immune, Toll-like receptors, cytokines, avian β-defensins

## Abstract

**Simple Summary:**

Cold stress has an adverse effect on immune, antioxidant, and neuroendocrine systems as well as growth performance of broilers. Much attention was focused on how to improve the cold resistance of animals. However, it is not clear how intermittent and mild cold stimulation regulate the immune function. Therefore, the present study was designed to investigate the effect of intermittent and mild cold stimulation on the immune function of bursa in broilers. Our results indicated that the intermittent and mild cold stimulation (IMCS) could help broilers adapt to the low ambient temperature and maintain homeostasis by modulating the production of Toll-like receptors (TLRs), cytokines, and avian β-defensins (AvBDs) in the bursa of broilers. We expect to find a way to promote the establishment of cold adaptation through repeated intermittent cold trainings.

**Abstract:**

Cold stress causes growth performance to decrease and increases production costs. Cold adaptation can enhance immune function and alleviate the negative impact caused by the stress condition. The study investigated the effect of intermittent and mild cold stimulation on the immune function of the bursa of Fabricius in broilers. A total of 400 healthy one-day-old broilers were divided into the control group (CC) and cold stimulation (CS) groups. The CC group was raised at a conventional raising temperature of broilers, while the CS groups were raised at 3°C below the temperature of the CC for three-, four-, five-, or six-hour periods at one-day intervals from 15 to 35 days of age (D35), denoted CS3, CS4, CS5, and CS6, respectively. Subsequently, they were raised at 20°C from 36 to 49 days of age (D49). The expression levels of TLRs, cytokines, and AvBDs were determined to access the immune function of bursa in broilers. After 21-day IMCS (at D36), the expression levels of TLR1, TLR15 and TLR21, interleukin (IL)-8, and interferon (IFN)-γ, as well as AvBD8 in CS groups, were lower than those in CC (*p* < 0.05). The expression levels of TLR3, TLR4 and TLR7, were decreased in the CS3, CS5, and CS6 groups (*p* < 0.05), but there were no significant differences in both the CC and CS4 groups (*p* > 0.05). When the IMCS ended for 14 days (at D49), the expression levels of TLR2, TLR3, TLR5, TLR7, TLR15, and TLR21, and IL-8, as well as AvBD2, AvBD4 and AvBD7 in CS groups, were lower than those in CC (*p* < 0.05). In addition to CS4, the expression levels of TLR1, IFN-γ, and AvBD8 in CS3, CS5, and CS6 were still lower than those in CC (*p* < 0.05). We concluded that the intermittent and mild cold stimulation could regulate immunoreaction by modulating the production of TLRs, cytokines, and AvBDs in the bursa, which could help broilers adapt to low ambient temperature and maintain homeostasis.

## 1. Introduction

In the alpine region, livestock and poultry often suffer from cold stress due to the low ambient temperature in winter. Cold stress has adverse effects on animal growth performance as well as antioxidant, immune, and neuroendocrine systems [1,2,3,4,5]. Studies have shown that the organism’s ability to adapt to the environment could be improved if the body accepted a variety of adverse environmental stimuli during the growth period. For example, long-term exposure of livestock and poultry to an environment slightly below rearing temperature could improve cold tolerance of the body and prevent and mitigate cold damage to a certain extent [6]. Cold acclimation can enhance the immune function and disease resistance of the body. Our previous study has shown that when broilers were continuously given a cold stimulation of 3 °C lower than rearing temperature for the period of 8 to 42 days of old, they were more likely to adapt to ambient temperature changes during their later growth stages [7]. Li et al. [8] found that early cold stimulation at 3 °C below the control group temperature could improve the antioxidant function in broilers. Cell-mediated immune function was enhanced in mice after two weeks of cold adaptation at 2 °C [9]. Moderate cold stimulation will not destroy homeostasis in the early growth of broilers, but will enhance disease resistance in later growth stages [10,11,12,13].

The bursa of Fabricius, the unique and primary humoral immune organ, plays a crucial role in B-lymphocyte proliferation and differentiation in chicken [14]. A large number of differentiated B-lymphocytes, migrating from the bursa of Fabricius to peripheral lymphoid organs for colonization and reproduction, are responsible for executing important immune functions. Toll-like receptors, which are involved in diverse recognition of highly conserved pathogen-associated molecular patterns (PAMPs) [15], exhibit antiviral, antibacterial, antifungal, and antiparasitic biological activity in innate immunity. After binding to specific pathogen molecules, members of TLR families will induce signaling pathways and activate the expression of transcription factor, NF-κB, to trigger an inflammatory response or acquired immune response [16,17,18]. The binding of chicken TLR2 and TLR4 to lipoproteins of Gram-positive bacteria and lipopolysaccharide (LPS) of Gram-negative bacteria, respectively, directly induces inflammatory cytokines release from bursal cells [19,20,21]. Al-Zghoul et al. [22] observed that acute heat stress resulted in the significant upregulation of TLR4 mRNA expression in the spleen and liver of the broiler. TLR3 and TLR7 have been shown to be responsible for antiviral infection in the body [17,23]. Many studies in other species also discovered the effect of changes of ambient temperature stimulation on TLRs. Such as 21-day consecutively heat stress (35 °C) led to higher expression levels of TLR2 and TLR4 in peripheral blood mononuclear cell of Bama miniature pigs, and prolonged heat stress would be responsible for immunosuppression of Bama miniature pigs [24]. Moreover, it was found that the expression level of TLR2 in fish was increased with the decrease of ambient temperature after exposure to cold water (10, 15, and 20 °C) for seven days [25]. However, the effect of intermittent and mild cold stimulation on the gene expression levels of TLR family in broilers has not been reported so far.

Cytokines play a critical role in many biological processes and are known to be protein mediators [26]. Pattern recognition receptors, Toll-like receptors, can recognize pathogen-associated molecular patterns. Subsequently, the downstream cytokines, such as interleukins (IL-6, IL-8, and IL-17) and interferons (IFN-γ, IFN-α, and IFN-β), are induced through cellular signal pathways [27,28]. One study pointed out that once TLRs were activated by either pathogens or host-derived ligands, they would induce downstream signals that led to cytokines and chemokines production, thereby initiating inflammatory responses [29]. In chickens, cold stress resulting from rapidly decreasing the temperature from 21.7 °C to 10 °C was reported to not only enhance the expression of IL-4 and -6, but also reduce the expression level of IFN-γ [30]. Additionally, He et al. [31] observed an increase in IL-4 and IL-6 mRNA expressions in the spleens of broilers that suffered from heat stress (37 °C for eight hours/day) for 14 days. One study showed that chronic cold stress upregulated the mRNA expression levels of IL-17 in the duodena, jejuna, and ilea of broilers [32]. Study in rats indicated that seven-day cold exposure (4 °C) resulted in the suppression of IL-6 and IL-17 [33]. These researches suggested that some stress/stimulation can activate cytokines to regulate the immune system, and some stress condition would damage the immune systems of animals.

Avian β-defensins are one class of avian host defense peptides which are responsible for the first line of defense against microbial and bacterial invasion in innate and adaptive immune systems. In chickens, totally, 14 avian β-defensins have been identified from AvBD1 to AvBD14 [34]. Avian β-defensins are indispensable mediators of innate disease resistance in tissues. Disease infections such as *Salmonella* enterica serovar Enteritidis (SE) infection and necrotic enteritis (NE) lead to the upregulation of AvBD2, AvBD4, AvBD5, and AvBD8 in poultry [35,36]. It has been suggested that the production of AvBDs play a protective role against pathogen invasion. Thus, we expect to observe the immunomodulatory of AvBDs during our intermittent and mild cold stimulation experiment.

Therefore, the purpose of the present work was to explore whether the intermittent and mild cold stimulation could help broilers adapt to the low ambient temperature and maintain homeostasis modulating the expression levels of TLRs, cytokines, and AvBDs in the bursae of broilers.

## 2. Materials and Methods

### 2.1. Animal Care and Experimental Design

All experiments and procedures described in this study complied with the Guidelines of Northeast Agricultural University Rules Concerning Animal Care and Use and have been approved by the Northeast Agricultural University Animal Care and Use Committee (IACUCNEAU20150616). A total of 400 healthy one-day-old Ross-308 broilers were randomly divided into five equal groups: four treatment groups (CS3, CS4, CS5, and CS6) and a control group (CC), each with five replicates (16 chicks per replicate). The control group was kept at a conventional rearing temperature of broilers—that is, 35°C for the first 3 days, then reduced gradually by 1 °C every two days until reaching 20 °C at 33 days of age, and this temperature was maintained until 49 days of age (07:00). Cold stimulation groups CS3, CS4, CS5, and CS6 were kept at 3 °C lower than the temperature of CC for three-hour, four-hour, five-hour, and six-hour periods, respectively, at one-day intervals from 15 to 35 days of age; then, they were kept at 20 °C until 49 days of age (07:00).

Birds were reared in battery cages until slaughter. The cages were 180 cm × 80 cm × 60 cm. Birds had free access to food and water during the rearing period. A complete starter diet (21.00% of crude protein (CP); 12.10 mega joule (MJ)/kg of metabolizable energy (ME)) was given from 1 to 21 days of age of broilers; then, a grower diet (19.00% of CP; 12.80 MJ/kg of ME) was given from 22 to 42 days of age, and a finishing diet (17.50% of CP; 13.20 MJ/kg of ME) was provided until the end of the experiment. The composition of the diets included corn, soybean meal, vitamin A, vitamin D3, copper sulfate, calcium chloride, amino acid, mildew inhibitor, phytase, etc. We purchased the diets from Harbin Baishicheng Animal Husbandry Co. LTD. The relative humidity was maintained at 60–70%. The daily lighting regime was as 23 h light: 1 h dark.

### 2.2. Collecting Samples

During the experiment, two broilers from each replicate per group were randomly selected and euthanized at 08:00 am at 22, 29, 36, 43, and 49 days of age (namely, D22, D29, D36, D43, and D49), respectively. Subsequently, their bursa of Fabricius were quickly removed, and the collected tissue samples were immediately frozen in liquid nitrogen and stored at −80 °C until further use. All efforts were made to minimize the pain or discomfort of the chicks and to minimize the total number of chicks used in the study.

### 2.3. RNA Extraction and Quantitative RT PCR (qPCR) Analysis

Total RNA was extracted from the broilers’ bursa samples using an RNAiso Plus Kit (Takara, Dalian, China) according to manufacturer’s instructions. The quality of the RNA was checked using an ultra-microspectrophotometer (Gene Quant 1300/100, Boston, MA, USA). The corresponding cDNA was synthesized using a high-efficiency cDNA reverse transcription kit (Toyobo, Osaka, Japan) according to the manufacturer’s instructions. The cDNA was subsequently diluted with DEPC-treated water.

The primer sequences shown in Table 1 were designed using Primer Premier Software 5.0 (PREMIER Biosoft International, Palo Alto, CA, USA) and synthesized by Sangon Biotech Co. Ltd. (Shanghai, China). Quantitative real-time PCR was performed on a LightCycler^®^480 II (Roche, Switzerland). Each 10 µL reaction mixture contained 5 µL THUNDERBIRDSYBR qPCR Mix (Toyobo, Japan), 1 µL diluted cDNA, 0.3 µL of each primer (10 µM), and 3.4 µL PCR-grade water. The qPCR conditions were as follows: pre-denatured 90 °C for 60 s, followed by 40 cycles of denaturation at 95 °C for 15 s and annealing and extension steps at 60 °C for 1 min. The melting curve showed a single peak for each PCR product. The PCR reactions were performed in triplicate, and the threshold cycle (Ct) value used in subsequent calculations was the mean of the values from three reactions. The relative expression of each mRNA was calculated using the $2^{-\Delta \Delta \mathrm{CT}}$ method, and the housekeeping gene β-actin was used as an internal reference for normalization of the results.

### 2.4. Statistical Analysis

IBM SPSS Statistics 21 (IBM, Armonk, NY, USA) was used for statistical analysis. All data were tested for normal distribution using the Kolmogorov–Smirnov test. The differences among the CC and CS groups within the same sampling time and the differences among the sampling time within the same group were analyzed by one-way analysis of variance (ANOVA) with Tukey’s honest significant difference test. Data were expressed as mean ± standard deviation, and *p* < 0.05 was considered statistically significant.

## 3. Results

### 3.1. Relative Expression Levels of Toll-Like Receptors

Figure 1 shows the effect of intermittent cold stimulation on the mRNA expression of Toll-like receptors (TLR1, TLR2, TLR3, TLR4, TLR5, TLR7, TLR15, and TLR21) in the bursae of broilers. During intermittent and mild cold stimulation, the expression levels of most TLRs showed a trend of first increasing and then decreasing. The expression levels of TLR1, TLR7 and TLR15 in the CS groups, were significantly higher compared to those of the control group (*p* < 0.05) after 14-day IMCS (at D29). After 21-day IMCS (at D36), the expression levels of TLR1, TLR3, TLR4, TLR7, TLR15, and TLR21 in the CS3, CS5, and CS6 groups, and the expression levels of TLR1, TLR5, TLR15, and TLR21 in the CS4 group, were lower than those in the control group (*p* < 0.05). In addition, the expression levels of TLR1 and TLR15 in the CS5 group were lower compared to those of the CS3 and CS6 groups (*p* < 0.05). When the IMCS training ended for 14 days (at D49), the expression levels of all TLRs in both the CS5 and CS6 groups and the expression levels of TLRs except TLR1 and TLR4 in the CS3 and CS4 groups were significantly lower compared to those of the control group (*p* < 0.05). Among them, the expression levels of TLR1, TLR2, TLR3, and TLR5 in the CS5 group, were lower than those in the CS3 and CS4 groups (*p* < 0.05).

### 3.2. Relative Expression Levels of Cytokines

Figure 2 shows the effect of intermittent cold stimulation on the mRNA expression of cytokines (IL-6, IL-8, and IL-17 as well as IFN-γ, IFN-α, and IFN-β) in the bursae of broilers. The expression levels of IL-6, IL-17, IFN-γ, IFN-α, and IFN-β in CS groups except the CS6 group were significantly higher than those in the control group (*p* < 0.05) after seven-day IMCS (at D22). The expression levels of IFN-α and IFN-β in CS5 group were higher than those in other CS groups (*p* < 0.05). After 21-day IMCS (at D36), the expression levels of IL-8 and IFN-γ in the CS groups were lower compared to the control group (*p* < 0.05). The expression levels of IL-8 and IFN-γ in the CS5 group were lower than those in both the CS3 and CS4 groups (*p* < 0.05), and there was no significant difference in the expression level of IFN-γ between the CS5 group and the CS6 group (*p* > 0.05). When the IMCS training ended for 14 days (at D49), the expression levels of IL-6, IL-8, IL-17, and IFN-γ in the CS5 and CS6 groups were significantly higher than those in the control group (*p* < 0.05). Furthermore, the expression levels of IL-8 and IFN-α in the CS5 group were significantly lower than those in the CS6 group (*p* < 0.05); however, there were no significant differences in the expression levels of IL-6, IL-17, IFN-γ, and IFN-β between the CS5 group and the CS6 group (*p* > 0.05).

### 3.3. Relative Expression Levels of Avian β-Defensins (AvBDs)

Figure 3 shows the effect of intermittent cold stimulation on the mRNA expression of avian β-defensins (AvBD2, AvBD4, AvBD5, AvBD7, and AvBD8) in the bursae of broilers. The expression levels of AvBD2, AvBD4 and AvBD5 in the CS groups, were significantly higher than those in the control group (*p* < 0.05) after seven-day IMCS (at D22). After 21-day IMCS (at D36), the expression levels of AvBD2 and -8 in the CS5 group were significantly lower than those in the control group (*p* < 0.05), while there were no significant difference in the expression levels of AvBD4 and AvBD7 between the CS5 group and the control group (*p* > 0.05). Moreover, the expression levels of AvBD8 in the CS3, CS4, and CS6 groups were significantly lower than those in the control group (*p* < 0.05), while the expression levels of AVBD2, AvBD4, AvBD5, and AvBD7 in the CS4 and CS6 groups, had no significant difference from the control group (*p* > 0.05). Compared with the CS5 group, the expression levels of AvBD2 and AvBD8 in the CS4 and CS6 groups were significantly higher (*p* < 0.05). When the IMCS training ended for 14 days (at D49), the expression levels of AvBDs except AvBD5 and AvBD8 in the CS groups were significantly lower compared to the control group (*p* < 0.05). Furthermore, the expression levels of AVBD2, AVBD4 and AVBD7 in the CS5 group, were lower than those in both CS4 and CS6 groups (*p* < 0.05).

## 4. Discussion

It has been widely reported that cold stress can affect an organism’s normal functioning, especially regarding immune function [4,32]. Currently, researchers have found that proper cold exposure would not have an adverse effect on animals but would improve their adaptability to a low-temperature environment. For example, Hangalapura et al. [37] demonstrated that broilers’ humoral immune function was improved after a seven-day cold exposure. In previous studies, Su [7] and Wei [13] demonstrated that broilers housed at a temperature of 3°C lower than ambient temperature could enhance their immunomodulatory and antioxidative functions when subjected to acute cold stress. Therefore, our study was designed to further examine the effect of intermittent mild cold stimulation on the bursae of broilers by characterizing the mRNA expression of TLRs, cytokines, and AvBDs.

The recognition and binding of pathogen-associated molecule pattern (PAMP) proteins by Toll-like receptors triggers a series of signaling cascades which activate related cytokines and play an important role in innate immune response [17,18,38]. Al-Zghoul, Saleh, and Ababneh [22] found that TLR4 mRNA expression level was upregulated in the spleens and livers of broilers under acute heat stress. The expression of TLR2 in the spleens of broilers was significantly upregulated by short-term acute heat stress with a rapid increase of 16°C for 10 h [39]. Zhang et al. [40] found that TLR4 in the spleens of triploid carp was rapidly elevated after being infected by *Aeromonas hydrophilia* (a Gram-negative bacterium) for six hours. The above studies demonstrated that the stress condition or the invasion of bacteria could affect immune function of broilers, and that they upregulated the mRNA expression levels of TLRs. We found that the expression levels of most TLRs were lower in the CS groups compared to the control group after 21-day IMCS, with the lowest expression levels of TLR1, TLR3, TLR4, TLR5, TLR7, and TLR15 in the CS5 group. These results suggested that the IMCS training could regulate the expression levels of TLRs in the bursae of broilers and keep it in a stable state with low expression levels. Similarly, previous work of Quinteiro-Filho et al. [41] indicated that chronic thermal stimulation (10 h per day for six days) prior to sampling reduced TLR2 expression in the spleens of broilers. In this study, when the IMCS training ended for 14 days, most TLRs were still significantly lower in the CS groups than those in CC. It was obvious that the expression of TLRs were still at low levels, though the intermittent and cold stimulation had ended for 14 days. We speculated that broilers had gradually adapted to the low-temperature environment after repeated IMCS training; thus, the expression levels of TLRs changed from the high expression levels at the early stage of IMCS to the low expression levels at the later stage of IMCS. We also found that the expression levels of TLRs in CS5 group were lower and more stable. These results indicated that intermittent and mild cold stimulation can effectively regulate the expression levels of TLRs in the bursae of broilers and ensure the stability of immune function of broilers.

Cytokines are characterized by mediating immune reactions and being involved in inflammatory responses [22,28]. Our results showed that CS3, CS4, and CS5 groups had significantly upregulated the expression levels of IL-6, IL-17, IFN-γ, IFN-α, and IFN-β at the early stage of IMCS. A possible reason is that at the early stages of IMCS, a sudden change of ambient temperature would have an adverse effect on broilers, which would induce the body to produce pro-inflammatory cytokines. Our results are consistent with some other studies. Brenner et al. [42] found that the plasma levels of IL-6 in human were increased during cold exposure with prior heating and exercise. In response to cold stress, the IL-6 level of serum was enhanced in Wistar rats [43]. One study reported that the release of IFN-γ was enhanced with mice treated by mild pain training for 21 days [44]. Additionally, the expression of IL-6 could mitigate the damage from cold stress [45]. According to Bagchi et al. [46], changes from the external environment can lead to an increase in pro-inflammatory cytokines. The research from Akihisa Harada et al. [47] indicated that IL-8 played a causative role in acute inflammation by recruiting and activating neutrophils. In our study, after 21-day IMCS, the expression levels of IL-8 and IFN-γ were downregulated. From here, we can see that with the repeated IMCS training, broilers had shown a trend of adaptation to the low temperature, and the expression level of IL-8 was downregulated by the IMCS. When the IMCS training ended for 14 days, IL-6, IL-8, and IL-17 in some CS groups still kept low levels and were significantly downregulated. Among them, IL-17 in the CS5 and CS6 groups were significantly lower than that in other groups. Similar to IL-8, IL-17 is a class of pro-inflammatory cytokines [48,49]. After the intermittent and mild cold stimulation ended, the low expression level of IL-17 was also a manifestation of cold adaptation in broilers. Simultaneously, the expression level of IL-8 in the CS5 group was significantly lower than those in the other three CS groups, and thus, the pro-inflammatory cytokines in the bursa of broilers in CS5 group were in a better state. Taken together, intermittent and mild cold stimulation could help broilers to reach cold adaptation, and the state could be maintained with the IMCS training ended.

Avian β-defensins, a subclass of antimicrobial peptides, play an important role in the innate immune system. Modulation of avian β-defensins may be one way to protect and improve animal health [34]. In our present results, the upregulation of AvBD2, AvBD4, AvBD5, and AvBD8 in the CS groups at the early stage of cold stimulation were consistent with the finding of Albert van Dijk [50], who found that the expression levels of β-defensins would be increased after exposure to microbial pathogens or inflammatory factors. Li et al. [51] pointed out that cold exposure induced the expression of antimicrobial peptide in desert beetles. Similarly, Su [7] observed that one week of cold stimulation at 3°C lower than the control group induced an elevation in most of AvBDs in the trachea of broilers. The possible reason for the results in our study is that at the early stage of IMCS, low-temperature stimulation mobilized the immune system in the bursa of broilers and the broilers improved their defense capability by producing a large amount of β-defensins in response to the sudden changes of ambient temperature. In this study, most of AvBDs in both of the CS4 and CS6 groups had no significant differences, but the expression level of AvBD8 was decreased. We found that most of the β-defensin genes (AvBD2, AvBD4, AvBD7, and AvBD8) in the CS groups still kept lower levels compared to the control group after the IMCS training ended for 14 days. These results indicated that intermittent and mild cold stimulation training enable broilers to adapt to the low ambient temperature and maintain a stable state with β-defensin genes in a low expression levels.

## 5. Conclusions

Based on the results of the present study, the intermittent and mild cold stimulation of 3°C below the conventional feeding temperature for three-, four-, five-, or six-hour periods at one-day intervals at an early stage could easily activate the innate immune system in the bursae of broilers and induce the production of Toll-like receptors, cytokines, and avian β-defensins to protect the body against damage. As the intermittent and mild cold stimulation prolonged, some TLRs, pro-inflammatory cytokines, and AvBDs were downregulated. It demonstrated that broilers finally established the cold adaptation by 21-day intermittent and mild cold stimulation, which ensured the stable state (homeostasis) in the bursae of broilers. Furthermore, when the IMCS ended for 14 days, some immune-related molecules still maintained the low expression levels, which indicated that the adaptability to the low-temperature environment was sustainable.

## Figures and Tables

**Figure 1 animals-10-01275-f001:**
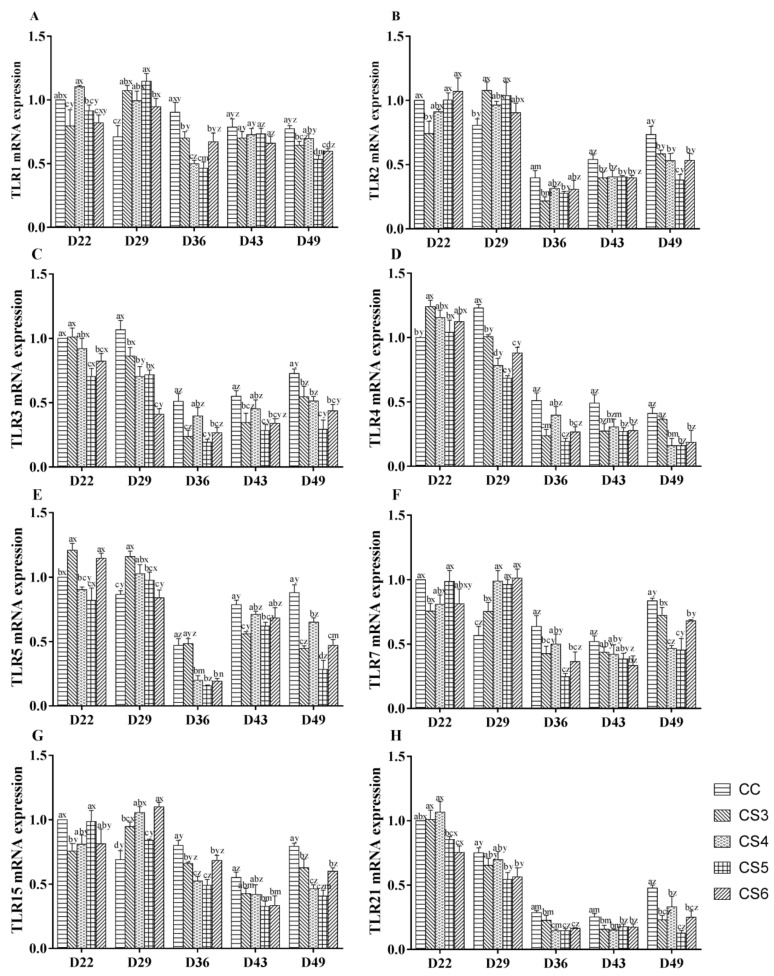
The mRNA expression levels of TLR1 (**A**), TLR2 (**B**), TLR3 (**C**), TLR4 (**D**), TLR5 (**E**), TLR7 (**F**), TLR15 (**G**), and TLR21 (**H**) in the bursae of broilers at D22, D29, D36, D43, and D49 (where D denotes “days of age” and the numeral indicates the day number, e.g., D22 refers to 22 days of age). a, b, c, d, and e represent significant differences among groups at the same sampling time (*p* < 0.05). x, y, z, m, and n represent significant differences among different sampling time in the same group (*p* < 0.05). CS3, CS4, CS5, and CS6 refered to the intermittent and mild cold stimulation of 3°C below the conventional feeding temperature for three-, four-, five-, or six-hour periods at one-day intervals, respectively.

**Figure 2 animals-10-01275-f002:**
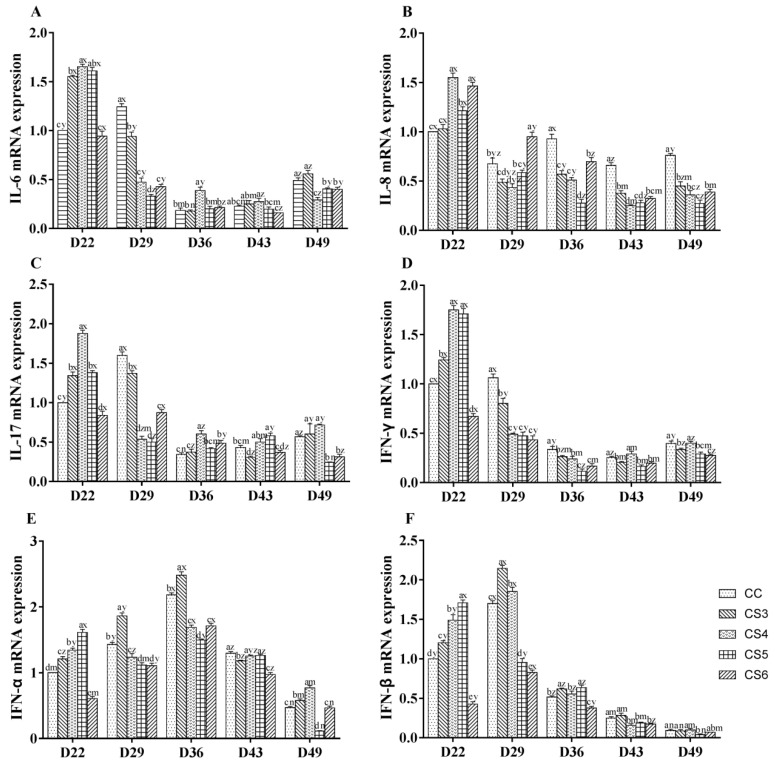
The mRNA expression levels of IL-6 (**A**), IL-8 (**B**), IL-17 (**C**), IFN-γ (**D**), IFN-α (**E**), and IFN-β (**F**) in broiler bursae of Fabricius at D22, D29, D36, D43, and D49 (where D denotes “days of age” and the numeral indicates the day number, e.g., D22 refers to 22 days of age). a, b, c, d, and e represent significant differences among groups at the same sampling time (*p* < 0.05). x, y, z, m, and n represent significant differences among different sampling time in the same group (*p* < 0.05). CS3, CS4, CS5, and CS6 refered to the intermittent and mild cold stimulation of 3°C below the conventional feeding temperature for three-, four-, five-, or six-hour periods at one-day intervals, respectively.

**Figure 3 animals-10-01275-f003:**
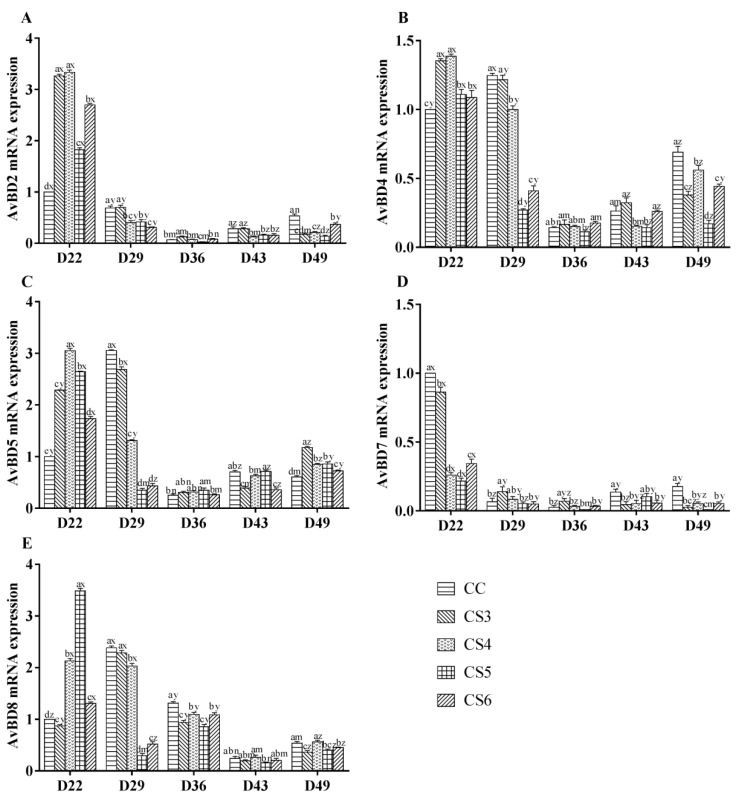
The mRNA expression levels of AvBD2 (**A**), AvBD4 (**B**), AvBD5 (**C**), AvBD7 (**D**), and AvBD8 (**E**) in broiler bursae of Fabricius at D22, D29, D36, D43, and D49 (where D denotes “days of age” and the numeral indicates the day number, e.g., D22 refers to 22 days of age). a, b, c, d, and e represent significant differences among groups at the same sampling time (*p* < 0.05). x, y, z, m, and n represent significant differences among different sampling time in the same group (*p* < 0.05). CS3, CS4, CS5, and CS6 refered to the intermittent and mild cold stimulation of 3°C below the conventional feeding temperature for three-, four-, five-, or six-hour periods at one-day intervals, respectively.

**Table 1 animals-10-01275-t001:** Gene primer sequences used for qPCR analysis.

Gene	Reference Sequence	Primer Sequences (5′-3′)
TLR1	NM_001081709	Forward: AGTCCATCTTTGTGTTGTCGCC Reverse: ATTGGCTCCAGCAAGATCAGG
TLR2	XM_001232192	Forward: GATTGTGGACAACATCATTGACTC Reverse: AGAGCTGCTTTCAAGTTTTCCC
TLR3	NM_001011691	Forward: TCAGTACATTTGTAACACCCCGCC Reverse: GGCGTCATAATCAAACACTCC
TLR4	NM_001030693.1	Forward: AGTCTGAAATTGCTGAGCTCAAAT Reverse: GCGACGTTAAGCCATGGAAG
TLR5	NM_001024586	Forward: CCTTGTGCTTTGAGGAACGAGA Reverse: CACCCATCTTTGAGAAACTGCC
TLR7	NM_001011688	Forward: TTCTGGCCACAGATGTGACC Reverse: CCTTCAACTTGGCAGTGCAG
TLR15	NM_001037835	Forward: GTTCTCTCTCCCAGTTTTGTAAATAGC Reverse: GTGGTTCATTGGTTGTTTTTAGGAC
TLR21	NM_001030558	Forward: TGCCCCTCCCACTGCTGTCCACT Reverse: AAAGGTGCCTTGACATCCT
IL-6	NM_204628.1	Forward: AAATCCCTCCTCGCCAATCT Reverse: CCCTCACGGTCTTCTCCATAAA
IL-8	NM_205018.1	Forward: GGCTTGCTAGGGGAAATGA Reverse: AGCTGACTCTGACTAGGAAACTGT
IFN-γ	NM_205149.1	Forward: GAACTGGACAGGGAGAAATGAGA Reverse: ACGCCATCAGGAAGGTTGTT
IFN-α	XM_004937097.1	Forward: GGACATGGCTCCCACACTAC Reverse: GGCTGCTGAGGATTTTGAAGA
IFN-β	NM_001024836.1	Forward: CACCACCACCTTCTCCT Reverse: TGTGCGGTCAATCCAGT
AvBD2	NM_204992	Forward: GGTTGTCTTCGCCCCGGCGGGA Reverse: TTATGCATTCCAAGGCCATTTG
AvBD4	NM_001001610	Forward: TCATCGTGCTCCTCTTTGTG Reverse: AATACTTGGGACGGCATAGC
AvBD5	NM_001001608	Forward: GCTGTCCCTTGCTCGAGGATT Reverse: GGAATACCATCGGCTCCGGC
AvBD7	NM_001001194	Forward: ACCTGCTGCTGTCTGTCCTC Reverse: TGCACAGCAAGAGCCTATT
AvBD8	NM_001001781	Forward: TTCTCCTCACTGTGCTCCAA Reverse: AAGGCTCTGGTATGGAGGTG
β-actin	NM_205518.1	Forward: CACCACAGCCGAGAGAGAAAT Reverse: TGACCATCAGGGAGTTCATAGC

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
