# Peer review of "Effect of Intermittent and Mild Cold Stimulation on the Immune Function of Bursa in Broilers"

_animals, 2020, doi:10.3390/ani10081275_

Round 1

Reviewer 1 Report

Thank you for your hard work and great job, The manuscript right now is much better and in acceptable form 

Author Response

Thank you very much for your review!

Reviewer 2 Report

Paper is an interesting topic relevant to the journal.  

Author Response

Thank you very much for your review!

Reviewer 3 Report

The Authors described the effect of intermittent and mild cold stimulation on the immune function of bursa in broilers. The experiment was carried out on Ross-308 broiler chickens from day one of life to 49 days of life.

I wonder why until 49 day of life? Why Authors used only starter and finisher diets but not grower diet? Also the composition of diets should be added to the Manuscript. If the diets were commercial the name of diets/manufacturer should be given. Why the light/dark cycle was 23h light : 1h dark but not 12:12 hours?

The hypothesis of the study should be given not at “Conclusions” section but at the end of the Introduction. Also the need of such studies should be better justify. Why is the need of such observations?

Figures are practically illegible and it is difficult to evaluate the correctness of the data.

Some fragments of “Discussion” are like section “Results”. It should be rewritten (e.g. l.252-262).

Reviewer 4 Report

This is a well-thought and formulated study, even though at 15 to 35 days broilers have accumulated most of their adipose tissue and can utilize heat increment better. At 3 ℃ below the 20 ℃, which may not be a challenge in tropical regions  during this period of production may likely have an effect in cold climate region especially with over-time application (15 to 35 days), but on a day or two, the effects may not be pronounced.

Does the expression of these immune defense agents not protect the host from microbial infection? Do low expressions reflect a good or bad prognosis as far as pathogenic infections are concerned? So if, the expression of innate and adaptive immunity is reduced, doesn't that reflect that these agents are not as alert as they should be? Is it possible to articulate further on this. It is a pity that production data was not a focus in this study, because that would reflect BW gains, feed intake, FCRs as well as mortality rates which are more critical quantitative measures of performance by poultry producers, or food/farm animal producers in general.

Author Response

This manuscript is a resubmission of an earlier submission. The following is a list of the peer review reports and author responses from that submission.

Round 1

Reviewer 1 Report

Dear Authors,

Thank you for submitting this research paper.  unforttunately, I have several issues with the paper, especially the results presentation, due to which I stopped proofreading and reviewing the paper. 

Simple summary: there are several missing words (like in line 13 "growth", line 17 "intermittent" etc) and a strong need for rephrasing many sentences (e.g. line 13 following, line 17 following).

Summary: the summary is unfortunately unprecise and the result-section missunderstandably written (e.g. regarding the weeks 3, 4 and 5/ also regarding "following cold stimulation"...)

Introduction: You use several abbreviation without introducing them. That's not applicable. And some point, you are unprecise like in line 72. Also Salmonella Enteritidis should be written correctly!

Material and methods: here, you are also using some abbreviations without introducing them. Also, more information about the statistical software is missing. 

Results: This part is written horrible and many of the statements are wrong compared to the diagram.  I'm sorry to say this, but you have to rewrite it. 

This is also the reason, why I stopped reviewing the paper.

Kind regards

Author Response

Dear Professor, 

Reviewer 2 Report

see the attached copy. The study is well designed, with an adequate number of replications, and demonstrated applicable results of values

Mean differences need more strick test rather than Duncan. 

The conclusion needs an improvement to indicate the time of implementation and duration of implementation. 

I suggest corrections as indicated in the attachment before acceptance 

Author Response

Dear Professor,

Reviewer 3 Report

Overall paper needs editing for English.  Line 14 is incomplete sentence.  Lines 73,83 also require attention.

Figures as presented are hard to interpret.

The idea is that intermittent cold stress will rescue the impact of cold stress on the immune system. The data presented don’t show that.

Line 218   States “The present study showed that the mRNA expression levels of TLR1, -2, -5, -7, and -15 were significantly elevated in contrast to the control group after two weeks of cold stimulation.”

But the experiment does not indicate that data was collected at 2 weeks, Furthermore, figure 1 does not show data for two weeks.  The data that is presented indicates that some TLRs are down regulated, some up regulated and some unchanged with intermittent cold stimulation.

Were birds checked for infection?  Can we be certain that the changes were due to environmental changes and not confounded by an infection or pathogen exposure.

Author Response

Dear Professor,
